# Report

# Preclinical validation of a novel metastasis-inhibiting Tie1 function-blocking antibody

Mahak Singhal[1,2,3,‡] (ID), Nicolas Gengenbacher[1,2,3,‡], Silvia La Porta[1,2,†,‡], Stephanie Gehrs[1,2,3], Jingjing Shi[1,2,3], Miki Kamiyama[1,2], Diane M Bodenmiller[4], Anthony Fischl[4], Benjamin Schieb[1,2], Eva Besemfelder[1], Sudhakar Chintharlapalli[4] & Hellmut G Augustin[1,2,5,*] (ID)

## Abstract

The angiopoietin (Ang)–Tie pathway has been intensely pursued as candidate second-generation anti-angiogenic target. While much of the translational work has focused on the ligand Ang2, the clinical efficacy of Ang2-targeting drugs is limited and failed to improve patient survival. In turn, the orphan receptor Tie1 remains therapeutically unexplored, although its endothelial-specific genetic deletion has previously been shown to result in a strong reduction in metastatic growth. Here, we report a novel Tie1 function-blocking antibody (AB-Tie1-39), which suppressed postnatal retinal angiogenesis. During primary tumor growth, neoadjuvant administration of AB-Tie1-39 strongly impeded systemic metastasis. Furthermore, the administration of AB-Tie1-39 in a perioperative therapeutic window led to a significant survival advantage as compared to control-IgG-treated mice. Additional *in vivo* experimental metastasis and *in vitro* transmigration assays concurrently revealed that AB-Tie1-39 treatment suppressed tumor cell extravasation at secondary sites. Taken together, the data phenocopy previous genetic work in endothelial Tie1 KO mice and thereby validate AB-Tie1-39 as a Tie1 function-blocking antibody. The study establishes Tie1 as a therapeutic target for metastasis in a perioperative or neoadjuvant setting.

**Keywords** angiogenesis; angiopoietin–Tie signaling; cancer; endothelial cells; metastasis

**Subject Categories** Cancer; Immunology; Vascular Biology & Angiogenesis

See also: **KA Khan & RS Kerbel** (June 2020)

## Introduction

Anti-angiogenic drugs targeting the VEGF-VEGF receptor pathway have been clinically approved more than 10 years ago. In fact, anti-angiogenesis marked the first clinically effective anti-stroma tumor therapy leading to an average increase in overall survival (OS) in different solid tumors of approximately 25% (Apte *et al*, 2019). Yet, this gain translates in absolute numbers in an increase in OS of only weeks to months (Ferrara & Adamis, 2016). The limited efficacy of clinically approved anti-angiogenic drugs has stimulated intense research in industry and academia to identify and validate second-generation anti-angiogenic targets that would either combine with anti-VEGF/VEGFR drugs or substitute for anti-VEGF/VEGFR drugs in patients with anti-VEGF/VEGFR non-responsiveness or resistance (Carmeliet & Jain, 2011; Jayson *et al*, 2016; Kuczynski *et al*, 2019).

Among the most intensely pursued second-generation anti-angiogenic candidate molecules is the contextual agonistic and antagonistic Tie2 ligand angiopoietin-2 (Ang2) (Saharinen *et al*, 2017). Similar to VEGF, Ang2 is prominently upregulated in essentially all types of solid tumors (Huang *et al*, 2010; Saharinen *et al*, 2017). Yet, unlike VEGF, which is predominately expressed and secreted by tumor cells, Ang2 is in almost all tumors produced by the tumor-associated angiogenic endothelial cells (ECs) and not by the tumor cells (Augustin *et al*, 2009; Helfrich *et al*, 2009). Ang2 thereby acts autocrine to regulate EC responsiveness to multiple cytokines, including VEGF (Felcht *et al*, 2012; Benest *et al*, 2013). Preclinical experiments with Ang2 targeting, either as a monotherapy or in combination with anti-VEGF, have shown a transient delay in the growth of primary tumors with no major effects on mean metastatic burden (Nasarre *et al*, 2009; Srivastava *et al*, 2014). These preclinical observations could well relate to the fact that anti-Ang2 showed in human clinical trials some efficacy in progression-free survival (i.e., short-term effects), but not in OS (i.e., long-term effects) (Monk *et al*, 2014, 2016). Likewise, human clinical trials comparing anti-VEGF

1   Division of Vascular Oncology and Metastasis Research, German Cancer Research Center Heidelberg (DKFZ-ZMBH Alliance), Heidelberg, Germany
2   European Center for Angioscience (ECAS), Medical Faculty Mannheim, Heidelberg University, Heidelberg, Germany
3   Faculty of Biosciences, Heidelberg University, Heidelberg, Germany
4   Eli Lilly and Company, Indianapolis, IN, USA
5   German Cancer Consortium, Heidelberg, Germany
    *Corresponding author. Tel: +49 6221 421500; E-mail: augustin@angiogenese.de
    †Present address: Springer Nature, Heidelberg, Germany
    ‡These authors contributed equally to this work

therapy with combined anti-VEGF/Ang2 combination therapy have not proven successful so far (Bendell et al, 2017).

In contrast to the well-understood Ang1/Ang2/Tie2 signaling axis, the functional role of the second Tie receptor, Tie1, during tumor progression and metastasis, remains elusive, which may mechanistically be largely due to its orphan receptor status (Eklund et al, 2017). Nevertheless, global genetic deletion of Tie1 results in late embryonic lethality as a consequence of perturbed vascular maturation, clearly showing critical and rate-limiting vascular functions of Tie1 (Puri et al, 1995; Sato et al, 1995). Correspondingly, the EC-specific postnatal deletion of Tie1 (Tie1$^{iECKO}$) leads to perturbed retinal angiogenesis with reduced vascular outgrowth and decreased numbers of outward growing tip cells (D'Amico et al, 2014; Savant et al, 2015). In a tumor context, Tie1 deletion affects intravasation of tumor cells at the primary tumor site and extravasation of metastasizing tumor cells at secondary sites, resulting in strongly reduced metastatic growth in Tie1$^{iECKO}$ mice (D'Amico et al, 2014; La Porta et al, 2018). Importantly, whereas Ang2 targeting affects primary tumor growth with no effect on later stages of tumor progression (Nasarre et al, 2009), tumor growth is only marginally affected in Tie1$^{iECKO}$ mice, but Tie1 targeting strongly suppresses metastasis (La Porta et al, 2018). This could suggest that Tie1 may target a clinically more relevant therapeutic window than Ang2. Moreover, the increasing appreciation that angiogenic regulators may directly affect metastasis, even independent of their angiogenic functions, warrants further study of second-generation anti-angiogenic targets as potential anti-metastatic drugs (Singhal & Augustin, 2020).

Building on the experiments in Tie1$^{iECKO}$ mice, the present study was aimed at generating and validating Tie1 function-blocking antibodies for therapeutic exploitation. This goal was complicated by the fact that the orphan receptor status of Tie1 and the limited understanding of the Tie1 mechanism of action made it difficult to establish a strategy for high-throughput screening to identify function-blocking antibodies. We therefore opted to screen for Tie1 antibodies that would inhibit Ang1-mediated Tie2 phosphorylation. These experiments resulted in the generation of antibody AB-Tie1-39, which in terms of postnatal retinal angiogenesis and metastatic growth phenocopied previous experiments in Tie1$^{iECKO}$ mice. AB-Tie1-39 thereby fully corroborated these earlier genetic experiments and validated Tie1 as a promising therapeutic target for further translational exploitation.

# Results and Discussion

### Generation and validation of a Tie1 function-blocking antibody

Monoclonal antibodies were raised against the extracellular domain of human Tie1. Six human Fab phage display libraries (de Haard et al, 1999) (FL323-03, FL323-05, FL323x, FL323xx, FL169-04, and FL169x) were toward this end panned against the extracellular domain of rhTie1 (rhTie1-ECD) antigen using the immunotube panning format. Three rounds of panning were carried out, and approximately 1,500 output-3 (O3) phages were screened for binding to biotin-labeled antigens by filter lift assay. Positive hits were then verified by DNA sequencing and assayed

by single-point ELISA (SPE) for binding to rhTie1-ECD. ELISA results identified 66 clones that bound to rhTie1-ECD. Of these, 22 Tie1-binding antibodies were screened for their effect on Ang1-stimulated Tie2 activation in human aortic ECs (Fig 1A). One antibody (AB-Tie1-39) robustly reduced AKT phosphorylation in an ELISA-based quantitation (Figs 1A and EV1A and B). Given that human and murine Tie1 share 92.62% sequence homology (Fig EV1C), AB-Tie1-39 displayed significant binding to murine Tie1 in a surface plasmon resonance-based conjugation assay (Fig EV1D).

During postnatal retinal angiogenesis, Tie1 contextually regulates Tie2 signaling as it counteracts Tie2 during active angiogenesis while sustaining its signaling in the remodeling plexus (Savant et al, 2015). To functionally validate AB-Tie1-39 in vivo, we studied post-natal retinal angiogenesis in AB-Tie1-39-treated pups. Newborn littermates were intraperitoneally injected with either AB-Tie1-39 or control IgG on postnatal days P2 and P4. Pups were sacrificed on P6, and isolated retinas were analyzed via high-resolution micro-scopy. The administration of AB-Tie1-39 suppressed retinal angiogenesis as evidenced by reduced vessel area and outgrowth (Fig EV2A and B). Further, a significant decline in the number of angiogenic tip cells was detected upon AB-Tie1-39 treatment (Fig EV2A and B). Concomitantly, an increased number of apoptotic EC were detected in the remodeling plexus of AB-Tie1-39-treated pups, marked by cleaved caspase 3 positivity (Fig EV2A and B). Collectively, AB-Tie1-39 not only restrained active angiogenesis, but also led to enhanced vessel regression during physiological vascular development.

### AB-Tie1-39 treatment delays primary tumor growth with no apparent vascular effects

Next, AB-Tie1-39 was administered to adult mice during different stages of tumor progression. To this end, preclinical metastasis models were employed, in which mice develop spontaneous multi-organ metastases after primary tumor resection (Srivastava et al, 2014; Gengenbacher et al, 2017). Modeling neoadjuvant therapy, mice were treated with either AB-Tie1-39 or control IgG during primary tumor growth (Figs 1B and EV3A). Two different primary tumor models (4T1 and Lewis lung carcinoma, LLC) coherently displayed a marginal reduction in primary tumor growth when mice were administered with AB-Tie1-39 as compared to IgG (Figs 1C and EV3B). These observations fully recapitulate previous experiments in a genetic model of EC-specific deletion of Tie1 (D'Amico et al, 2014; La Porta et al, 2018). However, unlike Tie1 deletion, AB-Tie1-39 treatment did not result in overt vascular changes as evidenced by no significant differences in intratumoral vessel density as well as perivascular coverage of tumor vessels when co-stained with pericyte (Desmin)- and smooth muscle cell (aSMA)-specific markers (Figs 1D and E, and EV3C and D). Additionally, tumors treated with AB-Tie1-39 did not show significant differences in tumor necrosis, vessel perfusion, and tumor hypoxia as compared to IgG-treated control mice (Figs 1E and EV3D and E). Collectively, the reduced tumor growth in two independent preclinical models despite no overt vascular changes and absence of tumor cell necrosis indicates that any anti-angiogenic effects of AB-Tie1-39 are rapidly compensated in a primary tumor context.

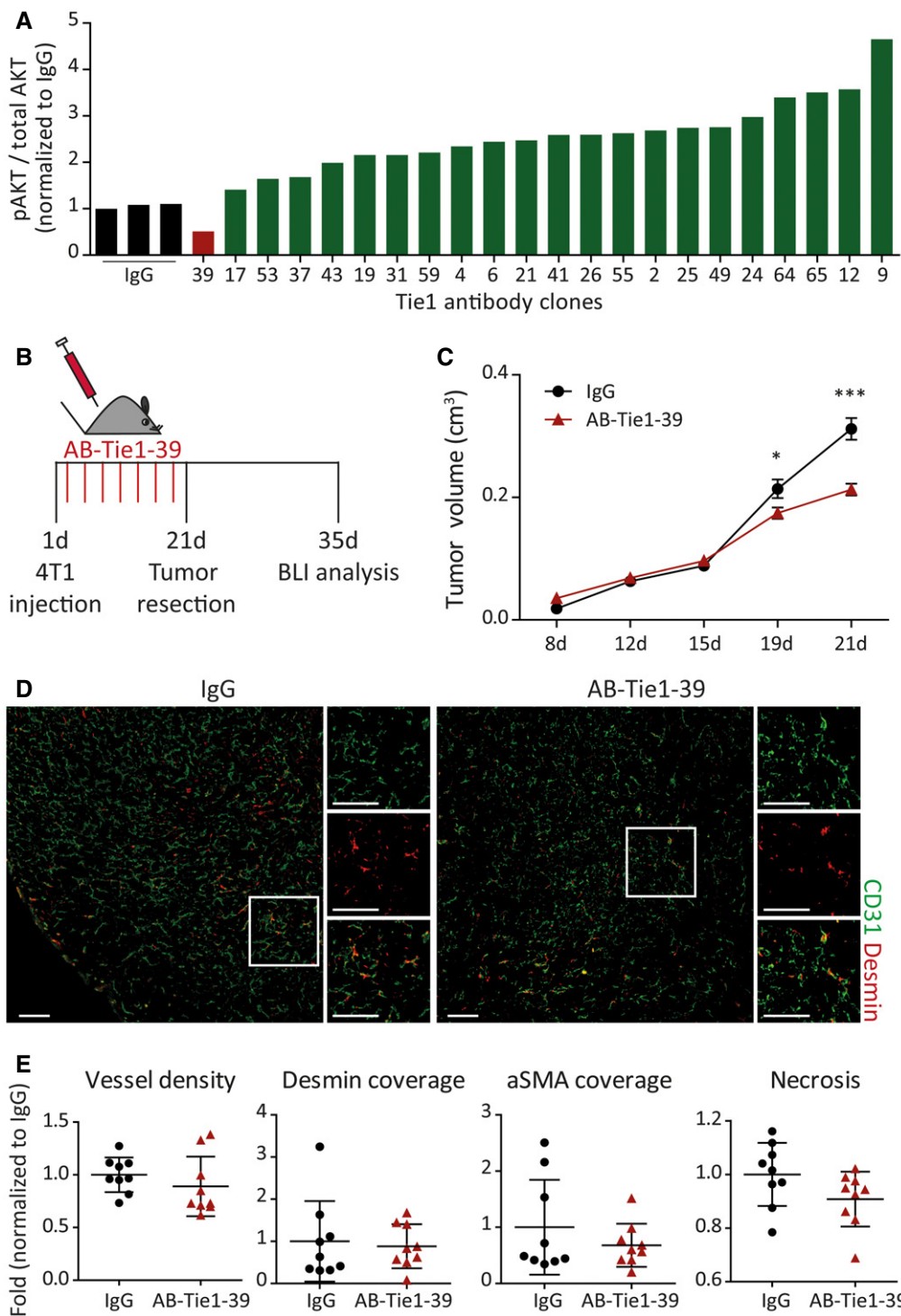

**Figure 1. Neoadjuvant treatment with AB-Tie1-39 slows tumor growth with no overt vascular defects.**

A   Twenty-two Tie1-binding antibodies were screened for phospho-AKT levels in an *in vitro* Ang1-stimulation experiment. Only one clone (AB-Tie1-39) reduced AKT phosphorylation indicating suppressed Tie2 signaling ($n = 1$ experiment).

B   Experimental outline of the spontaneous metastatic breast (4T1) cancer model treated with IgG or AB-Tie1-39 in a presurgical neoadjuvant setting.

C   Tumor growth curves show delayed primary tumor growth upon treatment with AB-Tie1-39 antibody as compared to IgG treatment (mean ± SEM, $n = 9$ mice). *, $P < 0.05$; ***, $P < 0.001$ (two-way ANOVA test).

D   Representative immunofluorescence images of tumor sections stained with CD31 (EC-specific marker) and Desmin (mural cell-specific marker). Scale bars = 200 μm.

E   Dot plots show quantitation of intratumoral vessel density, mural cell coverage using Desmin and aSMA co-staining, and tumor necrosis (mean ± SD, $n = 9$ mice). All comparisons were rendered non-significant according to two-tailed Mann–Whitney *U*-test.

## AB-Tie1-39 treatment restricts distant metastatic growth

Following neoadjuvant administration of AB-Tie1-39 or control IgG, primary tumors were resected to make metastasis rate-limiting for tumor progression. Ectopically luciferase-expressing 4T1 mouse mammary tumor cells enabled the non-invasive monitoring of metastatic progression by whole-body bioluminescence imaging (BLI). Mice treated with AB-Tie1-39 showed a profound reduction in distant metastases with a majority of mice displaying no detectable bioluminescence signal (Fig 2A and B). Subsequently, individual metastatic organs were subjected to *ex vivo* BLI. In full concordance with whole-body BLI, AB-Tie1-39-administered mice had a dramatic

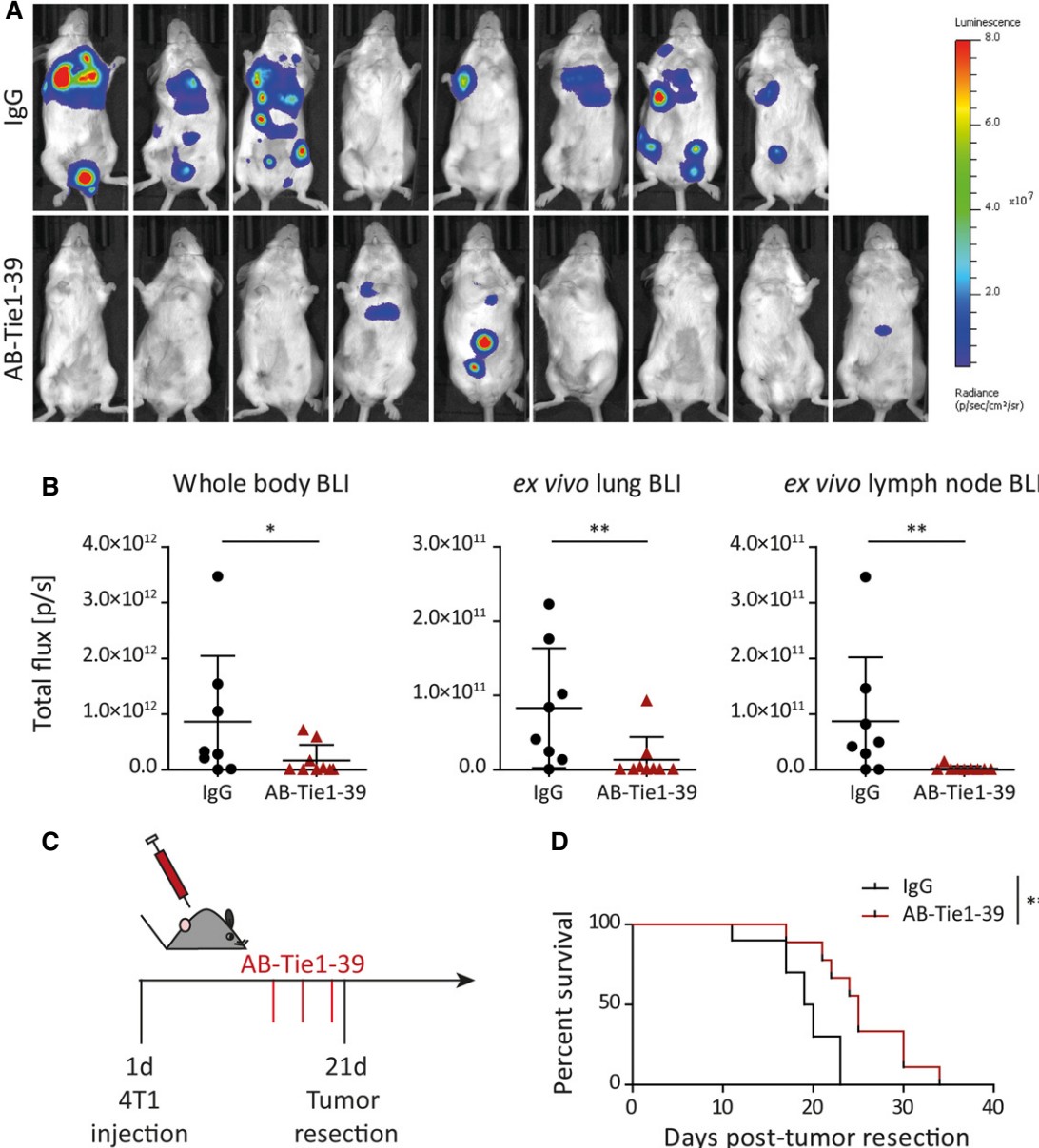

**Figure 2. Treatment with AB-Tie1-39 inhibits metastasis in the 4T1 breast cancer model.**

A   Following a neoadjuvant treatment regimen with IgG or AB-Tie1-39, mice were monitored by whole-body bioluminescence imaging. Shown are the images acquired 2 weeks after primary tumor resection.

B   Dot plots quantifying total photon flux during *in vivo* whole body as well as *ex vivo* lung and lymph node bioluminescence imaging (mean ± SD, $n_{IgG}$ = 8, $n_{AB-Tie1-39}$ = 9 mice). *$P$ < 0.05; **$P$ < 0.01 (two-tailed Mann–Whitney $U$-test).

C   Experimental outline of spontaneous metastatic breast (4T1) cancer model treated with IgG or AB-Tie1-39 in a perioperative setting. Therapy was initiated once tumors had reached an average size of 150 mm³.

D   Kaplan–Meier graph comparing percent survival between mice treated with either IgG or AB-Tie1-39 ($n_{IgG}$ = 10, $n_{AB-Tie1-39}$ = 9 mice). **$P$ < 0.01 (log-rank (Mantel–Cox) test).

reduction in tumor cell colonization in both lung and lymph nodes (Figs 2B, and EV4A and B). To validate these findings in a second tumor model, we employed the LLC model. Neoadjuvant application of AB-Tie1-39 in LLC tumor-bearing mice led to a significant reduction in lung metastatic burden (Fig EV4C and D). Taken together, AB-Tie1-39 effectively inhibited metastatic progression when administered in a neoadjuvant therapeutic regimen.

In order to test the anti-metastatic efficacy of AB-Tie1-39 in a clinically more realistic setting, a perioperative therapeutic strategy was employed. Here, AB-Tie1-39 treatment was initiated at an advanced stage of primary tumor growth, when tumors had reached an average tumor volume of around 150 mm$^3$, and therapy was terminated after primary tumor resection (Fig 2C). Unlike neoadjuvant treatment, perioperative administration of AB-Tie1-39 in 4T1 tumor-bearing mice did not limit primary tumor growth compared with IgG-treated mice (Fig EV4E). Nevertheless, late AB-Tie1-39 intervention still resulted in a significant survival advantage over IgG-treated animals (Fig 2D). These data indicate that the reduction in metastasis observed upon neoadjuvant AB-Tie1-39 therapy was not a mere consequence of primary tumor growth reduction and

suggest that AB-Tie1-39 treatment affected later steps of the metastatic cascade such as tumor cell seeding and/or colonization.

We next analyzed whether AB-Tie1-39 treatment affected metastatic colonization by only administering AB-Tie1-39 or IgG after primary tumor (4T1 or LLC) resection, thereby mimicking an adjuvant therapeutic regimen. Postsurgical treatment with AB-Tie1-39 yielded no survival advantage in the 4T1 model (Fig EV4F) and very similar metastatic burden to IgG-treated mice in the LLC model (Fig EV4G). The lack of efficacy of adjuvant AB-Tie1-39 administration negated a major role of Tie1 during colonization of tumor cells.

To assess the effect of AB-Tie1-39 treatment on seeding and extravasation of circulating tumor cells at a distant metastatic site, wild-type mice were preconditioned with either AB-Tie1-39 or IgG for 1 week. Thereafter, melanoma (B16F10) cells were intravenously injected to mimic metastatic progression independent of the primary tumor (Fig 3A). Mice were sacrificed after 2 weeks, and the number of lung metastatic foci was counted under a dissection microscope (Fig 3B). Mice pretreated with AB-Tie1-39 exhibited a strong reduction in lung metastases, thereby validating a contribution of Tie1 toward tumor cell seeding or extravasation at the metastatic site. To

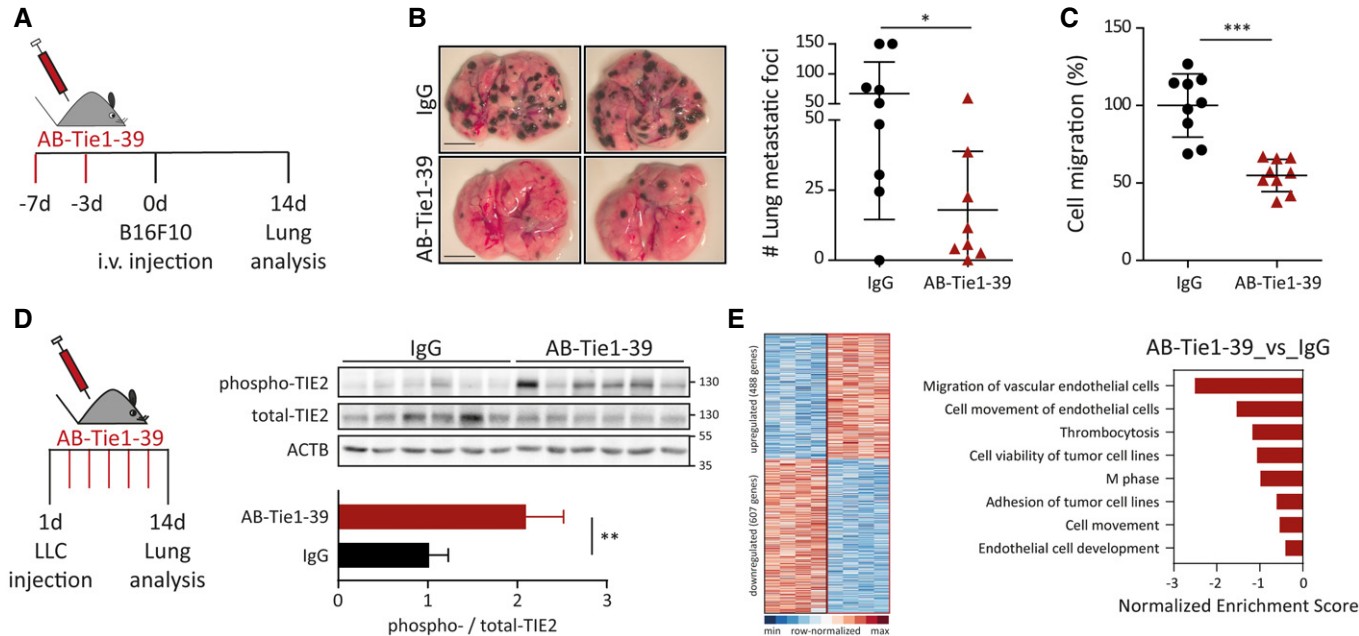

**Figure 3. AB-Tie1-39 treatment stabilizes lung vessels to restrict extravasation of disseminated tumor cells.**

A   Mice were pretreated with IgG or AB-Tie1-39. Thereafter, B16F10 cells were intravenously injected to initiate experimental metastasis independent of the primary tumor.

B   Representative images of the whole lungs are shown on the left. Scale bars = 5 mm. Quantitation of the number of lung metastatic foci is plotted on the right (mean ± SD, $n_{IgG}$ = 9, $n_{AB-Tie1-39}$ = 8 mice). *$P$ < 0.05 (two-tailed Mann–Whitney $U$-test).

C   ECs were seeded on gelatin-coated Transwell inserts. RFP-labeled LLC cells were allowed to transmigrate for 8 h through the endothelial layer. Thereafter, transmigrated tumor cells were counted. The dot plot presents the normalized data (mean ± SD, $n$ = 9 inserts from three independent experiments). ***$P$ < 0.001 (two-tailed Mann–Whitney $U$-test).

D   LLC tumor-bearing mice were pretreated with IgG or AB-Tie1-39. On day 14, lung tissue lysates were immunoblotted with anti-phospho-TIE2, total-TIE2, and ACTB (upper panel; $n$ = 6 mice). Densitometric quantitation of the blots presented in the upper panel is shown (mean ± SD, $n$ = 6 mice). **$P$ < 0.01 (two-tailed Mann–Whitney $U$-test).

E   LLC tumor-bearing mice were pretreated with IgG or AB-Tie1-39. On day 14, lung ECs were FACS-sorted and used for global gene expression ($n$ = 4 mice). On the left, a heatmap shows significantly regulated genes (red outline = AB-Tie1-39-treated; black outline = IgG-treated). On the right, top significantly regulated bio-functions in AB-Tie1-39-treated as compared to IgG-treated mice in ingenuity pathway analysis.

Source data are available online for this figure.

recapitulate the experimental metastasis assay in a more reductionist *in vitro* setting, human umbilical vein ECs (HUVECs) were seeded on gelatin-coated Transwell inserts. Upon forming a monolayer, HUVECs were treated with either AB-Tie1-39 or IgG. Thereafter, RFP-labeled LLC cells were allowed to transmigrate through the EC monolayer. Quantitation of transmigrated tumor cells demonstrated a robust decline when HUVECs were pretreated with AB-Tie1-39 as compared to IgG (Fig 3C). Collectively, the *in vivo* and the *in vitro* data suggest that Tie1 inhibition by AB-Tie1-39 blocked extravasation of disseminated tumor cells into a secondary organ.

### AB-Tie1-39 administration promotes vascular quiescence without altering the immune landscape in the metastasized lung

Genetic deletion of Tie1 induced vessel stabilization by inducing Tie2 phosphorylation (La Porta *et al*, 2018). Fully in line with genetic experiments, we found enhanced levels of Tie2 phosphorylation in lung lysates of AB-Tie1-39-treated as compared to IgG-treated mice (Fig 3D). Concurrently, global gene expression analysis of lung EC revealed a strong downregulation of gene sets corresponding to EC migration, cell development, and tumor cell adhesion in AB-Tie1-39-treated mice, indicating increased EC quiescence (Fig 3E). Notably, there were no changes in either total count or proliferation of lung EC upon AB-Tie1-39 treatment (Fig EV5A and B), suggesting that AB-Tie1-39 did not affect angiogenesis of lung EC, but promoted vascular quiescence by potentiating constitutive Tie2 signaling.

Immune cells represent an integral part of the metastatic microenvironment (Lim *et al*, 2018). In this context, shedding of the Tie1 ectodomain was previously reported to promote vascular remodeling and leakage during an acute inflammation (Kim *et al*, 2016; Korhonen *et al*, 2016). This prompted us to investigate whether AB-Tie1-39 would affect the immune milieu at the metastatic site. To this end, LLC tumor-bearing mice were treated with either AB-Tie1-39 or IgG. Thereafter, FACS-based immune phenotyping of the lung tissue was performed (Fig EV5A). There were no evident differences in the total number of leukocytes per mg of lung tissue (Fig EV5C). Likewise, no changes in the lymphoid nor in the analyzed myeloid cell populations were observed upon treatment with AB-Tie1-39 (Fig EV5D–F). Thus, despite a strong reduction in tumor cell extravasation, AB-Tie1-39 treatment did not alter the immune milieu within the metastatic niche.

In summary, employing different spontaneous preclinical metastasis models, the present study established and validated the Tie1 function-blocking antibody AB-Tie1-39 as a versatile tool for future Tie1 research and as a promising target for anti-metastatic therapy. AB-Tie1-39 perfectly recapitulated previously reported findings in genetic models that Tie1 can contextually act positively as well as negatively on Tie2 (Savant *et al*, 2015) (i.e., the AB was screened in cell culture for phospho-Tie2 inhibition, but acted *in vivo* on the resting lung vasculature in primary tumor-bearing mice in a phospho-Tie2-enhancing manner). Furthermore, the findings of this study demonstrated that (i) AB-Tie1-39 marginally delayed primary tumor growth without affecting the intratumoral vasculature; (ii) presurgical neoadjuvant administration of AB-Tie1-39 suppressed distant organ metastasis; (iii) AB-Tie1-39 selectively impeded extravasation of circulating tumor cells in the metastatic niche with no obvious effects on the composition of infiltrating immune cells; and (iv) short-term perioperative treatment with AB-Tie1-39, as a monotherapy, conferred a

significant survival advantage. In conclusion, by assessing different temporal therapeutic windows for intervention, the present study established the novel Tie1-binding antibody AB-Tie1-39 as a potent anti-metastatic agent, which warrants further translational investigation of Tie1 as a therapeutic target.

## Materials and Methods

### Study approval

All animal experiments were approved by the institutional and governmental Animal Care and Use Committees (G171/15, G231/16, G254/18, and G9/19 from Regierungspräsidium Karlsruhe, Germany). All experiments were performed in accordance with the institutional guidance for the care and use of laboratory animals.

### Mice

C57BL/6N and CB17-SCID mice were purchased from Charles River or Janvier Labs. Female mice (8–10 weeks of age) were used in this study unless otherwise indicated. All mice were housed on a 12-h light/dark cycle with free access to food and drinking water in specific pathogen-free animal facilities.

### Cells

Human umbilical vein ECs (PromoCell) were cultured in Endopan 3 media (PAN-Biotech). VeraVec human aortic ECs (HUAECs, hVer-a105) were cultured in endothelial basal media (Lonza) supplemented with SingleQuots growth factors (Lonza, CC-4147). LLC, 4T1-Luc2, and B16F10 cells (ATCC) were maintained according to ATCC standard culture instructions. RFP-labeled LLC (LLC-RFP) cells were kindly provided by Prof. Andreas Fischer (DKFZ, Heidelberg, Germany). All cells were cultured at 37°C and 5% $CO_2$ and routinely tested for mycoplasma by PCR.

### *In vitro* screening assay

Serum-starved HUAECs were pretreated with Tie1-binding antibodies (10 μg/ml) for 30 min. Thereafter, cells were stimulated with recombinant rhANG1 (R&D, 923-AN) at 300 ng/ml for 15 min. Cell lysates were collected, and phospho(Ser473)-AKT and total-AKT levels were analyzed using the MSD Cell Lysate Analysis Kit (K15100D). For phospho-Tie2 analysis, serum-starved HUVECs were pretreated with Tie1-binding antibodies (10 μg/ml) for 1 h. Thereafter, cells were stimulated with recombinant rhANG1 (R&D, 923-AN) at 400 ng/ml for 15 min. HUVEC lysates were analyzed using the human phospho-Tie-2 DuoSet IC ELISA Kit (DYC2720, RnD).

### Retinal angiogenesis assay

Newborn pups were administered with either AB-Tie1-39 or control IgG (40 mg/kg) intraperitoneally on postnatal days P2 and P4. On P6, eyeballs were fixed in methanol. Retinas were isolated and co-stained with FITC-conjugated isolectin B4 (IB-4, Sigma, L9381) and

cleaved caspase 3 (Cell Signaling, 9661). High-resolution 3D images were acquired on a Zeiss LSM 710 confocal microscope. Quantitation of staining intensities was performed with Fiji Is Just ImageJ (FIJI) software. For analysis of the tip cells, the number of tip cells (identified by their filopodia) at the retinal front was manually counted and normalized to the radial length of the vascular front.

## Tumor experiments

Three different tumor models were employed in this study (LLC, 4T1 breast cancer, and B16F10 melanoma). Mice in all tumor experiments were regularly monitored for ethical experimental endpoints.

## LLC tumor model

Lewis lung carcinoma cells ($1 \times 10^6$ in PBS) were inoculated subcutaneously in C57BL/6N mice. Primary tumors were surgically resected at an average size of 300 $mm^3$. Mice were administered with either AB-Tie1-39 or control IgG (40 mg/kg) twice a week. Therapy was initiated either 1 day after tumor cell implantation (neoadjuvant) or postprimary tumor resection (adjuvant). For quantitation of the metastatic score, randomized whole lung images were assigned a score by two independent researchers based on the reference images (Fig EV4D).

## 4T1 tumor model

4T1-Luc2 cells ($1 \times 10^5$ in PBS) were inoculated orthotopically in the fourth mammary pad of CB17-SCID mice. Primary tumors were surgically resected at an average size of 300 $mm^3$. Mice were administered with either AB-Tie1-39 or control IgG (40 mg/Kg) twice a week. Therapy was initiated either 1 day after tumor cell implantation (neoadjuvant) or at an average tumor volume of 150 $mm^3$ (perioperative). For postsurgical adjuvant therapy, treatment was initiated 1 day after primary tumor resection and continued until the end of the experiment.

## B16F10 experimental metastasis assay

C57BL/6N mice were pretreated with either AB-Tie1-39 or IgG (40 mg/Kg) twice for 1 week. Thereafter, B16F10 cells ($1 \times 10^5$ in PBS) were injected into the tail vein. Lungs were collected 2 weeks after tumor cell inoculation, and metastatic foci were counted under a stereo-microscope.

## Bioluminescence imaging and analysis

Following primary tumor resection, mice inoculated with 4T1-Luc2 cells were monitored weekly via *in vivo* bioluminescence imaging for metastatic growth. Briefly, mice were anesthetized with isoflurane and luciferase was recorded 10 min after injection of 2 mg luciferin with a Xenogen IVIS imaging system (Perkin Elmer). For photon flux analysis, the 4.4 live imaging software was used.

## Transmigration assay

HUVECs ($1 \times 10^5$) were plated in the top chamber of 6.5-mm/8.0-μm 0.2% gelatin-coated Transwells (Corning) overnight. The endothelial monolayer was pretreated with either AB-Tie1-39 or IgG (10 μg/ml) for 24 h. Thereafter, LLC-RFP cells ($1 \times 10^5$) were seeded in the top chamber in DMEM containing 10% FCS with DMEM containing 10% FCS also in the bottom chamber. Transwells were washed 8 h later and fixed with Roti Histofix (4% PFA) for 10 min. Transmigrated LLC-RFP cells were counted under a fluorescence microscope.

## Immunofluorescence stainings and analyses

Primary tumors were embedded in Tissue-Tek OCT compound and were cut into 5- to 7-μm sections. Tissue sections were fixed in ice-cold methanol and were blocked using 10% ready-to-use normal goat serum (Life Technologies, Thermo Fisher Scientific). The tissue sections were then incubated overnight at 4°C with primary antibodies [rat anti-CD31 (BD Biosciences, catalog 550300, 1:100); rabbit anti-Desmin (Abcam, catalog Ab15200-1, 1:200); and mouse anti-αSMA (Merck-Sigma, catalog C6198, 1:200)]. Staining with the secondary antibodies [anti-rat A488, anti-rabbit A546, and anti-rat A546 Abs (Life Technologies, Thermo Fisher Scientific, 1:500)] was performed next day for 1 h at room temperature. Cell nuclei were stained with Hoechst (Merck-Sigma). Images of whole tumor cross-sections were taken using a Zeiss Axio Scan slide scanner, and image analysis was performed using FIJI software.

## Intratumoral hypoxia and vessel perfusion analysis

Lewis lung carcinoma tumors were treated with AB-Tie1-39 or IgG. Hypoxyprobe (60 mg/kg) was injected intraperitoneally 1 h, and anti-CD31-PE (MEC13.3, 100 μl of 0.2 mg/ml) was injected intravenously 15 min prior to sacrificing mice. Primary tumors were embedded in Tissue-Tek OCT compound and were cut into 5- to 7-μm sections. Tissue sections were stained with Hypoxyprobe-Green Kit and anti-CD31 (AF3628, 1:100) antibody.

## FACS analysis of lung tissue

Lewis lung carcinoma tumor-bearing mice were treated with either AB-Tie1-39 or control IgG. Mice with an average tumor size of 300 $mm^3$ were euthanized, and lung tissue was collected. Tissue was dissociated into single-cell suspension with Liberase digestion enzyme mix (Roche). Following erythrocyte lysis, the remaining single-cell solution was divided for lymphoid [CD45 (30-F11, 1:400), CD3ε (17A2, 1:150), CD4 (GK1.5, 1:400), CD8a (53–6.7, 1:400), CD45R-B220 (RA3-6B2, 1:200), and NK-1.1 (PK136, 1:150)] and myeloid [CD45 (30-F11, 1:400), CD11b (M1/70, 1:200), Ly-6C (HK-1.4, 1:400), Ly-6G (1A8, 1:400), F4/80 (BM8, 1:100), MMR(C068C2, 1:200), and CD11c (N418, 1:400)] staining. Macrophage polarization analysis was performed as previously described (Kloepper *et al*, 2016). For EdU-based proliferation analysis, EdU (50 μg/g body weight) was intraperitoneally injected in LLC tumor-bearing mice. Mice were sacrificed 18 h later, and lungs were processed for FACS-based EdU analysis. Briefly, lung single-cell suspensions were stained with anti-CD45 (30-F11, 1:400) and anti-CD31 (MEC 13.3, 1:200) antibodies and processed according to the manufacturer's instructions using Click-iT EdU Pacific Blue Flow Cytometry Assay Kit (C10418, Thermo Fisher Scientific). Dead cells were excluded by FxCycle Violet staining. Stained cells were analyzed using a BD

Bioscience Aria Cell Sorting Platform, and frequency of individual cell populations was quantified with FlowJo software.

### Microarray analysis

For gene expression analysis, microarrays were performed by the German Cancer Research Center Genomics Core Facility. Briefly, lung ECs (DAPI$^-$ CD45$^-$ LYVE1$^-$ PDPN$^-$ TER119$^-$ CD31$^+$) were isolated from LLC tumor-bearing mice treated with either AB-Tie1-39 or control IgG. Thereafter, RNA was isolated with the Arcturus Pico-Pure RNA Isolation Kit (Life Technologies), and RNA quality and quantity were analyzed on an Agilent Bioanalyzer. Next, cDNA was hybridized on mouse Clariom S assay (Applied Biosystems) according to the manufacturer's protocol. Microarray data were normalized and analyzed with the Ingenuity Pathway Analysis software.

### Western blotting

Lewis lung carcinoma tumor-bearing mice were treated with either AB-Tie1-39 or IgG, and on day 14, lung tissues were snap-frozen for analysis. Subsequent to RIPA-based lysis, 50 μg tissue lysates were loaded on SDS–PAGE. Transferred blots were probed with anti-phospho-TIE2 (Y992; AF2720, R&D Systems, 1:1,000), anti-total-TIE2 (AF762, R&D Systems, 1:1,000), and anti-ACTB (sc-1616, Santa Cruz Biotechnology, 1:1,000). Densitometric quantification of pTIE2 and tTIE2 was performed using FIJI software.

### Statistical analysis

Statistical analysis was performed using GraphPad Prism version 6 (GraphPad Software). Data are expressed as the mean ± SD or SEM (as indicated). Comparisons between two groups were made using either a two-tailed Mann–Whitney *U*-test, two-way ANOVA, or log-rank (Mantel–Cox) test. A *P* value of < 0.05 was considered statistically significant. Mice were randomized before initiating therapy. For survival experiments, mice were daily observed by the animal caretakers who had no information about the biological groups. No animal experiments were repeated, and for *in vitro* experiments, the number of replicates is mentioned in the corresponding figure legends.

## Data availability

The microarray data generated in this study were deposited with the description to the Gene Expression Omnibus (GEO) repository and were made publically available under GEO accession no. GSE144851. (http://www.ncbi.nlm.nih.gov/geo/query/acc.cgi?acc = GSE144851).

**Expanded View** for this article is available online.

## Acknowledgements

The authors would like to thank Juqun Shen for performing immunization and phage panning for raising antibodies against Tie1. We are most grateful for the excellent technical support of the Flow Cytometry, Light Microscopy, and Laboratory Animal Facilities of the German Cancer Research Center. This work was supported by grants from the Deutsche Forschungsgemeinschaft

### The paper explained

**Problem**

Metastasis is the fatal hallmark of cancer. Interaction of seeded tumor cells with the local microenvironment, especially blood vessels, is crucial for successful metastatic colonization. Following the clinical success of VEGF/VEGFR-targeting drugs, the vascular cell-specific angiopoietin (Ang)–Tie signaling pathway has been pursued as a candidate for the development of second-generation anti-angiogenic therapy. While Ang2-targeting drugs have shown limited efficacy endothelial cell (EC)-specific deletion of the orphan receptor, Tie1 strongly suppressed distant organ metastases. However, due to a lack of specific antibodies, Tie1 remains hitherto unexplored as a druggable therapeutic target, which is also complicated by the orphan receptor status of Tie1.

**Results**

Here, we established and validated a novel monoclonal antibody (AB-Tie1-39) against human TIE1 which potently suppressed metastasis. We, further, evaluated the efficacy of AB-Tie1-39 in different clinically relevant therapeutic regimens and found that a short-term perioperative administration of AB-Tie1-39 could significantly improve overall survival as compared to control-IgG-treated mice. Mechanistically, treatment with AB-Tie1-39 restricted extravasation of circulating tumor cells at the metastatic site without affecting the local immune landscape.

**Impact**

Current anti-angiogenic therapies have limited clinical efficacy, which translates into an overall survival advantage in the range of a few weeks to months. There is an urgent need to identify novel druggable targets which can limit metastatic progression. Tie1 function-blocking antibody (AB-Tie1-39), by restricting the later steps of metastatic progression, can improve overall survival in spontaneous metastasizing mouse tumor models. AB-Tie1-39 will not only serve as a versatile tool for vascular research, but its preclinical anti-metastatic effects warrant further clinical investigation.

(project C5 within CRC1366 "*Vascular Control of Organ Function*" [project number 39404578 to H.G.A.] and project C3 within CRC-TR209 "*Liver Cancer*" [project number 314905040 to H.G.A.]); the European Research Council Advanced Grant "*AngioMature*" [project 787181 to H.G.A.]; and DFG-funded Research Training Group 2099 "*Hallmarks of Skin Cancer*" [project P8 to H.G.A.]. Miki Kamiyama received a fellowship from the Naito Foundation and the Nakatomi Foundation.

## Author contributions

MS, NG, SLP, SC, and HGA conceived and designed the study. AF, DMB, JS, MK, MS, NG, SG, and SLP performed most of the experiments. BS and EB provided technical support. MS, NG, SLP, and HGA analyzed and interpreted the data. HGA supervised the project. MS, NG, and HGA wrote the manuscript. All authors discussed the results and commented on the manuscript.

## Conflict of interest

The authors declare that they have no conflict of interest.

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
