## [Review Process File · EMBO Molecular Medicine]

Preclinical validation of a novel metastasis-inhibiting Tie1 function-blocking antibody

Mahak Singhal, Nicolas Gengenbacher, Silvia La Porta, Stephanie Gehrs, Jingjing Shi, Miki Kamiyama, Diane M Bodenmiller, Anthony Fischl, Benjamin Schieb, Eva Besemfelder, Sudhakar Chintharlapalli, Hellmut G Augustin

Review timeline:

Submission date:	15th Jul 2019
Editorial Decision:	3rd Sep 2019
Revision received:	11th Feb 2020
Editorial Decision:	13th Mar 2020
Revision received:	16th Mar 2020
Accepted:	18th Mar 2020

Editor: Céline Carret

Transaction Report:

1st Editorial

3rd Sep 2019

Thank you for the submission of your manuscript to EMBO Molecular Medicine. We have now heard back from the three referees whom we asked to evaluate your manuscript.

You will see that while the three referees find the study potentially interesting and important, issues are raised pertaining to the following points:

- mechanism of action of the antibody on Tie1 and Tie2 (ref. 2 and 3),
- characterisation of the antibody,
- mechanism on metastasis (must be tested in adjuvant therapy as well, as highlighted by all referees). In addition, clarifications, details and an enhanced discussion must be provided.

We would therefore welcome the submission of a revised version within three months for further consideration and would like to encourage you to address all the criticisms raised as suggested to improve conclusiveness and clarity. Please note that EMBO Molecular Medicine strongly supports a single round of revision and that, as acceptance or rejection of the manuscript will depend on another round of review, your responses should be as complete as possible.

***** Reviewer's comments *****

Referee #1 (Comments on Novelty/Model System for Author):

Please refer to the detailed review.

Referee #1 (Remarks for Author):

Background

This manuscript presents evidence for a new therapeutic - a tie1 receptor blocking antibody called AB-Tie-39 - as an effective drug to prevent or treat metastases, but not necessarily primary tumors. The rationale for the study is based mainly on prior tumor related studies involving tie1 postnatal knockout mice which suggested that tie1 - still an 'orphan' receptor - were less susceptible to tumor cell intravasation of potentially metastatic tumor cells at distant organ sites - thus resulting in reduced metastatic burden, with minimal apparent effects on primary tumor growth. The blocking antibody developed and used was based on screens designed to detect antibody with ability to inhibit ang-1-mediated tie1 phosphorylation. This antibody was found to blunt retinal angiogenesis and suppress in vivo metastatic tumor growth of the mouse Lewis lung carcinoma and the 4T1 breast cancer lines, whereas it only delayed primary tumor growth (and did so without any obvious vascular effects). Both presurgical (i.e., neoadjuvant) therapy studies as well as postsurgical (adjuvant) therapy studies were performed before and after primary tumor resection respectively. Finally, a limited immune profile analysis was undertaken, the results of which indicated little change detected in the "immune landscape" in lung metastases of antibody-treated mice.

Comments and Critique

This short communication is an excellent and interesting study. Aside from detailing a novel/new therapeutic, it looks like an example of a selective "anti-metastatic" agent rather than a more generic "anti-cancer" drug. This drug may prove to be clinically useful not only for cancer therapy - especially for adjuvant therapy of high-risk patients with early stage micrometastatic disease, but also for retinal-related vision loss diseases such as age-related macular degeneration. The paper is very clearly written and the experiments well controlled. I have a number of suggestions for minor revisions which could improve what is already a strong study:

1. The anti-Tie1 blocking Ab was selected by screening Abs that inhibit Ang1-mediated Tie2 phosphorylation. It is known that Ang1-Tie2 signalling mainly mediates maintaining vessel integrity and quiescence, and thus stabilizes newly formed blood vessels. AB-Tie1-39 selected by this way showed strongest inhibition of Ang1-Tie2 signalling, while the experiments summarized in this paper also showed AB-Tie1-39 strongly inhibited tumor cell intravasation and extravasation, preventing tumour cells migrating through EC layer. It appears that AB-Tie1-39 also promotes vessel integrity and stability. The authors should discuss this seemingly contradictory point concerning inhibition of Ang1-Tie2 signalling but promotion of vascular integrity. Given the assumed efficacy of TIE1-39 Ab in inhibiting metastasis and the evidence that targeting of ANG-2 has a positive effect against primary tumor growth-did the authors contemplate examining the combination of both approaches (targeting TIE1 and ANG2) in any of the tumor models discussed in the manuscript to examine whether such an approach can enhance overall survival. I'm not asking for such an experiment to be done, but some comment about it might make the paper more interesting.
2. Unlike the neoadjuvant treatment experiment, perioperative administration of AB-Tie1-39 in 4T1 tumor-bearing mice did not limit primary tumor growth compared to IgG-treated mice (Fig. EV4E). AB-Tie1-39 was given for 7 doses in the neo-adjuvant experiment (Fig.1B), whereas in the perioperative administration experiment AB-Tie1-39 Ab was only given for 3 doses; could this be the reason for lack of inhibition of 4T1 primary tumour growth?
3. In terms of a potential mechanism responsible for the effect of TIE1-39 Ab on metastatic disease, the authors suggest lack of immune-modulatory activity. Did the authors consider examining other mechanisms that might explain the assumed effect with respect to intravasation/extravasation, e.g. obvious mechanisms such as alterations in chemottractans, adhesion molecules or ECM degradation might be important options to examine. The use of techniques such as intravital videomicroscopy might be useful to confirm altered extravasation/intravasation as a result of TIE1-39 therapy. Again, I'm not requesting such experiments to be done for this paper.
4. Should the authors consider the combination of TIE1-39 with relevant chemotherapeutic approaches?
5. The authors examined LLC model in both neoadjuvant and adjuvant setting, however, 4T1 was only examined in terms of neoadjuvant approach. In the interest of completeness should they show data for adjuvant TIE1-39 for 4T1? Or comment on this question?

6. Page 12-authors suggest that TIE1-39 "inhibited advanced metastasis" yet the models used for examination (neoadjuvant and adjuvant) may more closely represent early metastatic disease. This should be clarified or modified.

Recommendation: Acceptable with minor revisions.

Referee #2 (Comments on Novelty/Model System for Author):

indeed, confirmation of experimental settings in at least two models is advised.

Referee #2 (Remarks for Author):

Summary

In the manuscript "Preclinical validation of a novel metastasis-inhibiting Tie1 function-blocking antibody", the authors describe a novel monoclonal antibody (AB-Tie1-39) against human TIE1 able to inhibit metastasis in murine metastatic models. They demonstrate the efficacy of this antibody as a potent anti-metastatic agent in presurgical neoadjuvant and short-term perioperative therapeutic regimens. The mechanism of action reported by the authors is the impairment of extravasation of circulating tumor cells in the metastatic niche, without the involvement of infiltrating immune cells.

Overall, the manuscript is well written and describes the discovery of a new promising anti-metastatic drug, however its mechanism of action is poorly characterized and discussed. The discussion is not mature enough and should be extended. Several conceptual issues somehow weaken the message conveyed by the authors, and those should be addressed.

Major concerns

In their previous work "La Porta et al, (2018) J Clin Invest 128: 834-845" the authors showed that EC-specific deletion of Tie1 strongly limits metastatic tumor cell extravasation and this decrease in metastatic dissemination is linked to reduced tumor vessel sprouting but improved vessel functionality. In this work, the Tie1- inhibition causes the impairment of extravasation of circulating tumor cells but does not show the same effects at the vascular level (no significant differences in intratumoral vessel density as well as perivascular coverage of tumor vessels): how do the authors explain these differences? This is the major discrepancy that is neither proved nor discussed in a convincing way. The fact that pharmacological inhibition of Tie1 does not recapitulate completely the effects displayed by EC-specific Tie1 ablation means that additional mechanisms linked to Tie1 inhibition in cells other than ECs should be considered. For instance, Tie1 and 2 are known to be expressed by macrophages (10.1371/journal.pone.0082088), so this should be carefully evaluated, in light of the lack of vascular effects following Tie-1 pharmacological inhibition. Although Tie-2 seems to be the prominent isoform expressed in in vitro polarized macrophages, Tie-1 is still present in macrophages (10.1371/journal.pone.0082088). Furthermore, in a TME context, the TAM Tie-2/Tie-1 ratio is not known and should be evaluated. Another option is that the extent of Ab accumulation in different tissue is different so that the K_i for the receptor can be reached only in certain cell types/compartments.

How can the pharmacological effects on endothelial cells and vessel organization justify the reduced extravasation? The in-vitro experiments on HUVEC cells are not sufficient to shed light on the mechanism. In AB-Tie1-39-treated pups, the authors found an increased number of apoptotic EC in retinas with consequent vessel regression. Why does this effect not occur in tumors? Could the vessel regression be possible in metastatic niche?

For the authors AB-Tie1-39 administration does not alter the immune landscape in the metastasized lung, but FACS analysis should be completed with M1/M2 macrophage polarization and T cells activation markers, and also performed at 35 days from LLC injection as well as in previous neoadjuvant treatments. This is connected with the first point.

Furthermore, the authors should better justify the reduced tumor growth following the treatments.

Why do they not consider the immune landscape in the primary tumor? The primary tumor microenvironment could be involved in this mechanism. Connected with the first point.

All these issues should be addressed by the authors.

Minor concerns

In some cases there is no correspondence with the numbering of the figures, (Figs. 2D and E) and (Figs. 4C and D) should be (Figs. 1D and E) and (Figs. EV4C and D) respectively. The authors should check and edit.

The sentence "tumors had reached an average tumor volume of around 150 mm³, and therapy was terminated after primary tumor resection (Fig. 2C)" is confusing based on the figure.

The authors should evaluate tumor hypoxic areas and vessel perfusion.

Some experiments performed on the LLC model (for example: adjuvant therapeutic regimen) should be also performed on the 4T1 model and vice versa.

Referee #3 (Comments on Novelty/Model System for Author):

The authors used multiple models, including a developmental angiogenesis model and multiple models of tumorigenesis and evaluation of the effects of the antibody, including orthotopic, subcutaneous, and iv injected, and the Ab was evaluated with different temporal delivery strategies. Potential improvements include testing of a dose response and effects of the antibody as adjuvant to chemotherapy, although these may not be necessary for this paper.

Referee #3 (Remarks for Author):

This is an interesting and important paper that reveals new insights into Tie1's enigmatic biology by demonstrating a role for Tie1 in metastasis by preventing circulating tumor cell extravasation. The results are important in that they show an effect of Tie1 signaling distinct from a global or endothelial cell-specific deletion of the gene. Although the exact mechanistic details at the level of endothelial cells in vivo remain to be elucidated, the data with respect to the effects on tumor growth and metastasis are solid and the conclusions are sound. However, there are questions about the effects of the antibody, particularly its specificity for Tie1 and the mechanisms of its effects, e.g., on Tie2 activation/phosphorylation and signaling and Tie1/Tie2 shedding, and these are detailed below.

Major Comments:

1. Mechanistic effects of AB-Tie1-39 - As the authors note, studying Tie1's function is complex in the absence of a ligand with which to stimulate the receptor, and because Tie1 modulates Tie2 activity, this was used to identify AB-Tie1-39. However, additional mechanistic data would be helpful. In this regard, the authors should show whether the Ab has an effect on Tie2 phosphorylation after ligand activation (Ang-1 and possibly Ang-2? Does it turn Ang-2 into an agonist?) and not just downstream Akt activation. Also, does the Ab have any effect on expression or shedding of Tie1 or Tie2 (at least in vitro) or on Tie1-Tie2 heterodimerization (which may be hard to show), which might explain its activity?
2. AB-Tie1-39 specificity - Presumably the Ab does not recognize Tie2, as their ECDs are rather divergent, but was this tested?
3. Effect of AB-Tie1-39 on EC apoptosis - Although the contextual effects of the Ab may be quite different in vitro vs. in vivo, does it have any effect on EC viability in ECs in vitro?
4. Adjuvant therapy - Have the authors tested the Ab as a true adjuvant to chemotherapy? Although this may be beyond the scope of the current manuscript, as the data demonstrate that the Ab clearly has independent effects, it might be interesting to know how much of an additive effect the Ab

confers.

Minor Comments:

1. Introduction, p. 3, paragraph 1 - The statement that, "... this gain translates in absolute numbers in an increase of OS within weeks to months" is confusing. "Within" implies that the improvement occurs quickly, whereas I believe the authors mean to say "an increase in OS of only weeks to months".
2. Fig. EV2A and B - Please define how tip cells were identified.
3. Dose response - How was the Ab dose chosen, and was a dose response tested?

1st Revision - authors' response

11th Feb 2020

Please see next page.

Response to Editor's and Reviewer's comments

Editor's comments

COMMENT 1: Thank you for the submission of your manuscript to EMBO Molecular Medicine. We have now heard back from the three referees whom we asked to evaluate your manuscript.

You will see that while the three referees find the study potentially interesting and important, issues are raised pertaining to the following points:

- mechanism of action of the antibody on Tie1 and Tie2 (ref. 2 and 3),
- characterization of the antibody,
- mechanism on metastasis (must be tested in adjuvant therapy as well, as highlighted by all referees). In addition, clarifications, details and an enhanced discussion must be provided.

We would therefore welcome the submission of a revised version within three months for further consideration and would like to encourage you to address all the criticisms raised as suggested to improve conclusiveness and clarity. Please note that EMBO Molecular Medicine strongly supports a single round of revision and that, as acceptance or rejection of the manuscript will depend on another round of review, your responses should be as complete as possible..

RESPONSE 1: We sincerely thank the editor for a very thoughtful judgment of the reviewers' comments. We have made a concerted effort to address all the critical comments of the reviewers. We summarize the key findings which we made in revising the manuscript:

- AB-Tie1-39 was produced by screening for reduction of Tie2 phosphorylation and subsequent Akt phosphorylation in *in vitro*-cultured confluent monolayers of human endothelial cells (manuscript Figs. 1A and EV1A and B). Intriguingly though, administration of AB-Tie1-39 in tumor experiments *in vivo* resulted in enhanced TIE2 phosphorylation (manuscript Fig. 3D). These apparently contradictory data truthfully recapitulate the findings from previous genetic work, in which Tie1 was found to contextually regulate Tie2 negatively and positively (La Porta *et al.*, 2018, Savant *et al.*, 2015). It can therefore be inferred from the data that the different effects of AB-Tie1-39 on confluent cultured HUVEC (likely mimicking remodeling stalk cells) and on resting lung EC in primary tumor bearing mice reflect different contextual states of Tie1 vs. Tie2 on EC. Together, these data show that AB-Tie1-39 induces *in vivo* a Tie2 gain-of-function phenotype, thereby leading to vascular stabilization in the lung and impeding extravasation of circulating tumor cells (as also corroborated by gene expression analysis [see below]).
- To better characterize and understand the molecular mechanism underlying the effect of AB-Tie1-39, we performed a global gene expression analysis of lung endothelial cell (EC) isolated from LLC tumor-bearing mice (manuscript Fig. 3E). We show that neoadjuvant administration of AB-Tie1-39 led to a significant downregulation of gene sets related to EC migration and development indicative of enhanced vessel maturation. Additionally, gene sets related to adhesion of tumor cells were downregulated, fully substantiating our observed phenotype of reduced extravasation in both *in vivo* experimental metastasis and *in vitro* transmigration assays (manuscript Fig. 3A-C). Overall, AB-Tie1-39 confers a strong anti-migrastatic effect to limit distant metastases.
- During revisions, we have performed a detailed characterization of AB-Tie1-39. First, we found that AB-Tie1-39 specifically binds to Tie1, not to Tie2 (Fig. 4 of response to reviewers). Second, AB-Tie1-39 did not alter the gene expression of either Tie1 or Tie2 (both *in vivo* and *in vitro*) and did not impact the shedding of Tie1 and Tie2 (Fig. 3 of response to reviewers). Third, AB-Tie1-39 promoted vascular stabilization without affecting EC viability (Fig. 5 of response to reviewers, manuscript Fig. EV5A and B).
- We included in the revised manuscript postsurgical adjuvant treatment experiments in the 4T1 metastasis model (manuscript Fig. EV4F). Additionally, we show that combining a maximum-tolerated dose of chemotherapy (Abraxane, PTX) with AB-Tie1-39 did not yield significant improvement in mouse survival post-tumor resections as compared to AB-Tie1-39 treatment alone (Fig. 1 of response to reviewers).

To conveniently trace the additional experiments incorporated into the revised manuscript, all new data are marked in blue. Moreover, we have, as suggested by the reviewers, extended the discussion. However, the stringent length restrictions limited an elaborate discussion of all parts of the manuscript. Overall, the revised manuscript incorporates all the essential data to address the reviewers' specific comments, which – we believe – has very much helped to further improve the manuscript.

COMMENT 2: EMBO Molecular Medicine has a "scooping protection" policy, whereby similar findings that are published by others during review or revision are not a criterion for rejection. Should you decide to submit a revised version, I do ask that you get in touch after three months if you have not completed it, to update us on the status.

RESPONSE 2: We have fully adhered to the formatting guidelines for a Report in EMBO Molecular Medicine.

Reviewer #1

GENERAL COMMENT: This manuscript presents evidence for a new therapeutic - a tie1 receptor blocking antibody called AB-Tie-39 - as an effective drug to prevent or treat metastases, but not necessarily primary tumors. The rationale for the study is based mainly on prior tumor related studies involving tie1 postnatal knockout mice which suggested that tie1 - still an 'orphan' receptor - were less susceptible to tumor cell intravasation of potentially metastatic tumor cells at distant organ sites - thus resulting in reduced metastatic burden, with minimal apparent effects on primary tumor growth. The blocking antibody developed and used was based on screens designed to detect antibody with ability to inhibit ang-1-mediated tie1 phosphorylation. This antibody was found to blunt retinal angiogenesis and suppress *in vivo* metastatic tumor growth of the mouse Lewis lung carcinoma and the 4T1 breast cancer lines, whereas it only delayed primary tumor growth (and did so without any obvious vascular effects). Both presurgical (i.e., neoadjuvant) therapy studies as well as postsurgical (adjuvant) therapy studies were performed before and after primary tumor resection respectively. Finally, a limited immune profile analysis was undertaken, the results of which indicated little change detected in the "immune landscape" in lung metastases of antibody treated mice.

Comments and Critique

This short communication is an excellent and interesting study. Aside from detailing a novel/new therapeutic, it looks like an example of a selective "anti-metastatic" agent rather than a more generic "anti-cancer" drug. This drug may prove to be clinically useful not only for cancer therapy - especially for adjuvant therapy of high risk patients with early stage micrometastatic disease, but also for retinal-related vision loss diseases such as age related macular degeneration. The paper is very clearly written and the experiments well controlled. I have a number of suggestions for minor revisions which could improve what is already a strong study:

RESPONSE TO GENERAL COMMENT: We sincerely appreciate the reviewer's positive assessment of the manuscript. A concerted effort was made to address all the critical comments of the reviewers, which – we believe – has very much helped to further advance the manuscript.

COMMENT 1: The anti-Tie1 blocking Ab was selected by screening Abs that inhibit Ang1-mediated Tie2 phosphorylation. It is known that Ang1-Tie2 signaling mainly mediates maintaining vessel integrity and quiescence, and thus stabilizes newly formed blood vessels. AB-Tie1-39 selected by this way showed strongest inhibition of Ang1-Tie2 signaling, while the experiments summarized in this paper also showed AB-Tie1-39 strongly inhibited tumor cell intravasation and extravasation, preventing tumor cells migrating through EC layer. It appears that AB-Tie1-39 also promotes vessel integrity and stability. The authors should discuss this seemingly contradictory point concerning inhibition of Ang1-Tie2 signaling but promotion of vascular integrity. Given the assumed efficacy of TIE1-39 Ab in inhibiting metastasis and the evidence that targeting of ANG-2 has a positive effect against primary tumor growth-did the authors contemplate examining the combination of both approaches (targeting TIE1 and ANG2) in any of the tumor models discussed in the manuscript to examine whether such an approach can enhance overall survival. I'm not asking for such an experiment to be done, but some comment about it might make the paper more interesting.

RESPONSE 1: We thank the reviewer for this very thoughtful comment. AB-Tie1-39 was screened based on its ability to inhibit Ang1-Tie2 signaling in *in vitro*-cultured human EC and it does so by reducing Tie2 phosphorylation and subsequent Akt phosphorylation (manuscript Figs. 1A and EV1A and B). *In vivo* administration of AB-Tie1-39 suppressed extravasation of tumor cells by promoting vessel integrity, which in the first instance sounds contradictory to the *in vitro* screening approach. However, a detailed analysis of the metastatic niche in mice receiving neoadjuvant treatment

of AB-Tie1-39 identified enhanced TIE2 phosphorylation (manuscript Fig. 3D). Further, global gene expression analysis revealed significant downregulation of bio-functions, regulating EC proliferation, migration, and development, in AB-Tie1-39 as compared to IgG treated mice (manuscript Fig. 3E). Together, these data indicate that AB-Tie1-39 truthfully recapitulates the findings from previous genetic work, in which Tie1 was found to contextually regulate Tie2 negatively and positively (La Porta *et al.*, 2018, Savant *et al.*, 2015). Together, these data show that AB-Tie1-39 induces *in vivo* a Tie2 gain-of-function phenotype, thereby leading to vascular stabilization in the lung and impeding extravasation of circulating tumor cells.

Given that Ang2 and Tie1 targeting agents affect different stages of tumor progression, we do agree with the reviewer that combining two agents might yield a synergistic benefit. However, considering that Tie1-targeting alone manifested a very strong anti-migrastatic anti-metastasis effect in the preclinical models with a near-complete abrogation of metastases in the 4T1 model, it would technically be challenging to assess any such combination therapies in preclinical models. It should be noted that published reports of combination therapies in preclinical models are most of the times based on titrated doses of each of the employed drug candidates, i.e., drugs are given at subsaturating doses. While this truthfully identifies therapeutic synergies, it in essence defeats the purpose by not showing that a second drug might have therapeutic benefit beyond the maximum therapeutic benefit of the other drug.

COMMENT 2: Unlike the neoadjuvant treatment experiment, perioperative administration of AB-Tie1-39 in 4T1 tumor-bearing mice did not limit primary tumor growth compared to IgG-treated mice (Fig. EV4E). AB-Tie1-39 was given for 7 doses in the neo-adjuvant experiment (Fig.1B), whereas in the perioperative administration experiment AB-Tie1-39 Ab was only given for 3 doses; could this be the reason for lack of inhibition of 4T1 primary tumor growth?

RESPONSE 2: It could well be the case that the short time course of the peri-operative experiment was responsible for the lack of inhibition. Yet, there is no way to circumvent this, because it was the deliberate intent of the perioperative window to target just a relatively short time window around surgery.

COMMENT 3: In terms of a potential mechanism responsible for the effect of TIE1-39 Ab on metastatic disease, the authors suggest lack of immune-modulatory activity. Did the authors consider examining other mechanisms that might explain the assumed effect with respect to intravasation/extravasation, e.g. obvious mechanisms such as alterations in chemoattractants, adhesion molecules or ECM degradation might be important options to examine. The use of techniques such as intravital video-microscopy might be useful to confirm altered extravasation/intravasation as a result of TIE1-39 therapy. Again, I'm not requesting such experiments to be done for this paper.

RESPONSE 3: To decipher potential molecular mechanisms for the effects of AB-Tie1-39, we have performed a detailed analysis of the lung metastatic niche. Apart from analyzing the cellular composition of the lung niche (manuscript Fig. EV5), we undertook global gene expression profiling of lung EC to unravel different adhesion and ECM molecules that might mediate extravasation of tumor cells. Ingenuity Pathway Analysis revealed that treatment with AB-Tie1-39 resulted in the downregulation of key gene sets corresponding to bio-functions such as migration of vascular endothelial cells and cell movement of endothelial cells (manuscript Fig. 3E), suggesting that AB-Tie1-39 promotes a quiescent phenotype in the lung endothelium. Additionally, we found a gene set corresponding to adhesion of tumor cell lines significantly downregulated in AB-Tie1-39-treated mice. Together, the data illustrate that AB-Tie1-39 stabilized the lung vasculature, thereby impeding the extravasation of disseminated tumor cells.

COMMENT 4: Should the authors consider the combination of TIE1-39 with relevant chemotherapeutic approaches?

RESPONSE 4: We thank the reviewer for bringing up a clinically-important issue. We combined neoadjuvant AB-Tie1-39 with a single dose of maximum-tolerated chemotherapy (Abraxane, PTX) in the 4T1 metastasis model. Considering the strong suppression of distant metastases upon AB-Tie1-39 treatment, we had anticipated that it might be technically challenging to assess additive or synergistic survival benefits in combinatorial approaches (see also response 1 above). Indeed, combining neoadjuvant AB-Tie1-39 with MTD chemotherapy failed to provide additive benefit for the survival of mice (Fig. 1). Interestingly, although not significant, the combination group did even slightly worse as compared to AB-Tie1-39 single arm, which suggests that potential toxicity of chemotherapy to the lung vasculature might have compromised the vessel stabilization effects of AB-Tie1-39. We plan to follow this finding up in future studies.

Figure 1. Combination of neoadjuvant AB-Tie1-39 and a single injection of Abraxane (PTX) in the 4T1 model. Kaplan-Meier plot shows the surviving fraction of mice following primary tumor resections (n = 7-9 mice). *, P<0.05, **, P<0.01, ***, P<0.001 (Log-rank (Mantel-Cox) test).

COMMENT 5: The authors examined LLC model in both neoadjuvant and adjuvant setting, however, 4T1 was only examined in terms of neoadjuvant approach. In the interest of completeness should they show data for adjuvant TIE1-39 for 4T1? Or comment on this question?.

RESPONSE 5: We performed postsurgical adjuvant treatment in the 4T1 model for the completeness of the manuscript (manuscript Fig. EV4F). Similar to the LLC model, adjuvant targeting of Tie1 did not yield any survival advantage in the 4T1 model.

COMMENT 6: Page 12-authors suggest that TIE1-39 "inhibited advanced metastasis" yet the models used for examination (neoadjuvant and adjuvant) may more closely represent early metastatic disease. This should be clarified or modified.

RESPONSE 6: We have modified the text to improve clarity and to avoid any possible misinterpretations.

Reviewer #2

GENERAL COMMENT: In the manuscript "Preclinical validation of a novel metastasis-inhibiting Tie1 function blocking antibody", the authors describe a novel monoclonal antibody (AB-Tie1-39) against human TIE1 able to inhibit metastasis in murine metastatic models. They demonstrate the efficacy of this antibody as a potent anti-metastatic agent in presurgical neoadjuvant and short-term perioperative therapeutic regimens. The mechanism of action reported by the authors is the impairment of extravasation of circulating tumor cells in the metastatic niche, without the involvement of infiltrating immune cells.

Overall, the manuscript is well written and describes the discovery of a new promising antimetastatic drug, however its mechanism of action is poorly characterized and discussed. The discussion is not mature enough and should be extended. Several conceptual issues somehow weaken the message conveyed by the authors, and those should be addressed.

RESPONSE TO GENERAL COMMENT: We sincerely appreciate the thorough and thoughtful review of our manuscript. A concerted effort was made to fully and wholeheartedly address all the specific comments of the reviewer. While revising the manuscript, we have performed global gene expression profiling of metastatic lung endothelial cells (manuscript Fig. 3E) and macrophage polarization analysis (manuscript Fig. EV5F) to add further mechanistic information. Additionally, we have elaborated the discussion part of the manuscript. We are most thankful for his/her constructive critique which very much helped to further advance the manuscript.

COMMENT 1: In their previous work "La Porta et al, (2018) J Clin Invest 128: 834-845" the authors showed that EC-specific deletion of Tie1 strongly limits metastatic tumor cell extravasation and this decrease in metastatic dissemination is linked to reduced tumor vessel sprouting but improved vessel functionality. In this work, the Tie1-

inhibition causes the impairment of extravasation of circulating tumor cells but does not show the same effects at the vascular level (no significant differences in intratumoral vessel density as well as perivascular coverage of tumor vessels): how do the authors explain these differences? This is the major discrepancy that is neither proved nor discussed in a convincing way. The fact that pharmacological inhibition of Tie1 does not recapitulate completely the effects displayed by EC-specific Tie1 ablation means that additional mechanisms linked to Tie1 inhibition in cells other than ECs should be considered. For instance, Tie1 and 2 are known to be expressed by macrophages (10.1371/journal.pone.0082088), so this should be carefully evaluated, in light of the lack of vascular effects following Tie-1 pharmacological inhibition. Although Tie-2 seems to be the prominent isoform expressed in *in vitro* polarized macrophages, Tie-1 is still present in macrophages (10.1371/journal.pone.0082088). Furthermore, in a TME context, the TAM Tie-2/Tie-1 ratio is not known and should be evaluated. Another option is that the extent of Ab accumulation in different tissue is different so that the K_i for the receptor can be reached only in certain cell types/compartments.

RESPONSE 1: We agree with the reviewer that the discrepancy between genetic and pharmacological targeting of Tie1 deserves further discussion. During retinal angiogenesis, genetic deletion of Tie1 resulted in approximately 40% reduction of the vascularized area as compared to WT pups (D'Amico *et al.*, 2014, Savant *et al.*, 2015). However, treatment with AB-Tie1-39 led to a modest 10% decrease in vessel area (manuscript Fig. EV2B), suggesting weaker anti-angiogenic potential of antibody-mediated targeting. In the context of tumors, EC-specific deletion of Tie1 resulted in reduced tumor vascular area, whereas we did not observe a major vascular phenotype in tumors treated with AB-Tie1-39. These apparent discrepancies could be due to different kinetics of genetic deletion versus pharmacological Tie1 targeting. Therefore, it is possible that early transient anti-angiogenic effects (translating into a minor tumor growth difference) of AB-Tie1-39 might have already been compensated at the time of our analysis.

We intentionally decided to not study any potential “transient” primary tumor effects, as subsequent perioperative treatment experiments (which did not affect primary tumor growth) and experimental metastasis assays (performed in the absence of a primary tumor) yielded comparable anti-metastatic phenotypes, clearly excluding the primary tumor as a dominant factor for the anti-metastatic efficacy of AB-Tie1-39.

We thank the reviewer for raising an important point concerning the expression of Tie1 and Tie2 on macrophages. We isolated circulating and tumor-infiltrating macrophages from LLC tumor-bearing mice. However, we failed to detect Tie1 expression on the isolated cells. The publication referred by the reviewer does show increased expression of Tie1 during inflammation. However, the authors of this paper employed *in vitro*-cultured human PBMC-derived monocytes and differentiated them into macrophages. It remains to be systematically evaluated whether subsets of mouse myeloid cells can express Tie1. We are following up on this in separate experiments, but evidence obtained thus far does not imply a major role of myeloid expressed Tie1.

Given that AB-Tie1-39 is a humanized TIE1-binding antibody, we performed Western blotting with LLC primary tumors and corresponding lung lysates and immunoblotted against human IgG (Fig. 2). Indeed, we saw less accumulation of AB-Tie1-39 in primary tumors as compared to the corresponding lung lysates which may explain the differential response of the antibody.

Figure 2. 10 μ g of tumor and lung lysates were loaded on an SDS-PAGE (n = 5 mice). Afterward, immunoblotting was performed against human IgG and Vinculin.

COMMENT 2: How can the pharmacological effects on endothelial cells and vessel organization justify the reduced extravasation? The *in-vitro* experiments on HUVEC cells are not sufficient to shed light on the mechanism. In AB-Tie1-39-treated pups, the authors found an increased number of apoptotic EC in retinas with consequent vessel regression. Why does this effect not occur in tumors? Could the vessel regression be possible in metastatic niche?

RESPONSE 2: The Angiopoietin-Tie pathway regulates vessel permeability. Constitutive Ang1-Tie2 signaling maintains vascular quiescence in adults (Saharinen *et al.*, 2017). We show in the present study that neoadjuvant treatment with

AB-Tie1-39 resulted in increased Tie2 phosphorylation (manuscript Fig. 3D), which is solidly established to enhance EC quiescence and promote vascular stabilization and maturation (Augustin et al., 2009, Savant et al., 2015). Furthermore, global gene expression analysis revealed significant downregulation of bio-functions, regulating EC proliferation, migration, and development, in AB-Tie1-39- as compared to IgG-treated mice (manuscript Fig. 3E), supporting antibody-mediated vascular stabilization.

To evaluate vessel regression in the metastatic niche, we quantitated the absolute number of lung EC per mg tissue by FACS. We did not observe changes in the total number of lung EC following treatment with AB-Tie1-39 (manuscript Fig. EV5A and B). Similarly, we did not observe changes in the proliferation of lung EC. Additionally, in the global gene expression profiling of lung EC, gene sets corresponding to apoptosis or cell death were not altered. Overall, the data support the concept that the treatment with AB-Tie1-39, via enhanced Tie2-signaling, promoted vascular stabilization and therefore, restricted extravasation of circulating tumor cells.

COMMENT 3: For the authors AB-Tie1-39 administration does not alter the immune landscape in the metastasized lung, but FACS analysis should be completed with M1/M2 macrophage polarization and T cells activation markers, and also performed at 35 days from LLC injection as well as in previous neoadjuvant treatments. This is connected with the first point.

RESPONSE 3: We have included the analysis of M1/M2 macrophage polarization in the revised manuscript (manuscript Fig. EV5F). We did not observe changes in the ratio of M1 to M2 macrophages.

There were no differences in the absolute numbers of infiltrating T- (either CD4+ or CD8+) cells with neoadjuvant AB-Tie1-39 treatment in the LLC model. Moreover, there was full recapitulation of the anti-metastatic phenotype in immunocompromised mice (SCID mice were employed for all 4T1 experiments). This rules out the involvement of T-cell activation in the observed phenotype.

COMMENT 4: Furthermore, the authors should better justify the reduced tumor growth following the treatments. Why do they not consider the immune landscape in the primary tumor? The primary tumor microenvironment could be involved in this mechanism. Connected with the first point.

RESPONSE 4: This question has already been addressed. Please see our response to comment 1.

COMMENT 5: In some cases, there is no correspondence with the numbering of the figures, (Figs. 2D and E) and (Figs. 4C and D) should be (Figs 1D and E) and (Figs. EV4C and D) respectively. The authors should check and edit.

RESPONSE 5: We sincerely apologize for our editorial mistakes. In revising the manuscript, we have tried intensely to eliminate all editorial mistakes.

COMMENT 6: The sentence "tumors had reached an average tumor volume of around 150 mm³, and therapy was terminated after primary tumor resection (Fig. 2C)" is confusing based on the figure.

RESPONSE 6: We have modified the figure and corresponding figure legend to avoid any confusion.

COMMENT 7: The authors should evaluate tumor hypoxic areas and vessel perfusion.

RESPONSE 7: We have included in the revised manuscript analyses of intratumoral hypoxia and vessel perfusion (manuscript Fig. EV3E).

COMMENT 8: Some experiments performed on the LLC model (for example: adjuvant therapeutic regimen) should be also performed on the 4T1 model and vice versa.

RESPONSE 8: For consistency, we have performed an adjuvant therapy experiment in the 4T1 metastasis model (manuscript Fig. EV4F).

Reviewer #3

GENERAL COMMENT: The authors used multiple models, including a developmental angiogenesis model and multiple models of tumorigenesis and evaluation of the effects of the antibody, including orthotopic, subcutaneous, and iv injected, and the Ab was evaluated with different temporal delivery strategies.

Potential improvements include testing of a dose response and effects of the antibody as adjuvant to chemotherapy, although these may not be necessary for this paper.

This is an interesting and important paper that reveals new insights into Tie1's enigmatic biology by demonstrating a role for Tie1 in metastasis by preventing circulating tumor cell extravasation. The results are important in that they show an effect of Tie1 signaling distinct from a global or endothelial cell-specific deletion of the gene. Although the exact mechanistic details at the level of endothelial cells *in vivo* remain to be elucidated, the data with respect to the effects on tumor growth and metastasis are solid and the conclusions are sound. However, there are questions about the effects of the antibody, particularly its specificity for Tie1 and the mechanisms of its effects, e.g., on Tie2 activation/phosphorylation and signaling and Tie1/Tie2 shedding, and these are detailed below.

RESPONSE TO GENERAL COMMENT: We sincerely appreciate the positive assessment of our manuscript. A concerted effort was made to fully and wholeheartedly address all the specific comments of the reviewer. We are most thankful for his/her constructive critique which very much helped to further advance the manuscript.

COMMENT 1: Mechanistic effects of AB-Tie1-39 - As the authors note, studying Tie1's function is complex in the absence of a ligand with which to stimulate the receptor, and because Tie1 modulates Tie2 activity, this was used to identify AB-Tie1-39. However, additional mechanistic data would be helpful. In this regard, the authors should show whether the Ab has an effect on Tie2 phosphorylation after ligand activation (Ang-1 and possibly Ang-2? Does it turn Ang-2 into an agonist?) and not just downstream Akt activation. Also, does the Ab have any effect on expression or shedding of Tie1 or Tie2 (at least *in vitro*) or on Tie1- Tie2 heterodimerization (which may be hard to show), which might explain its activity?

RESPONSE 1: We thank the reviewer for this very thoughtful comment. AB-Tie1-39 was screened based on its ability to inhibit Ang1-Tie2 signaling in *in vitro*-cultured human EC and it does so by reducing Tie2 phosphorylation and subsequent Akt phosphorylation (manuscript Figs. 1A and EV1A and B). *In vivo* administration of AB-Tie1-39 suppressed extravasation of tumor cells by promoting vessel integrity, which in the first instance sounds contradictory to the *in vitro* screening approach. However, a detailed analysis of the metastatic niche in mice receiving neoadjuvant treatment of AB-Tie1-39 identified enhanced TIE2 phosphorylation (manuscript Fig. 3D). Further, global gene expression analysis revealed significant downregulation of bio-functions, regulating EC proliferation, migration, and development, in AB-Tie1-39 as compared to IgG treated mice (manuscript Fig. 3E). Together, these data indicate that AB-Tie1-39 truthfully recapitulates the findings from previous genetic work, in which Tie1 was found to contextually regulate Tie2 negatively and positively (La Porta *et al.*, 2018, Savant *et al.*, 2015). Together, these data show that AB-Tie1-39 induces *in vivo* a Tie2 gain-of-function phenotype, thereby leading to vascular stabilization in the lung and impeding extravasation of circulating tumor cells.

Figure 3. A. Tie1 and Tie2 relative expression in lung EC with neoadjuvant treatment with AB-Tie1-39 or IgG. **B-D** HUVEC were treated with either IgG or AB-Tie1-39 (10 μ g/ml) for 24 h. Afterward, cells were lysed for analyzing gene expression of Tie1 and Tie2 (**B**). Additionally, condition media were used for performing immunoprecipitation of soluble either TIE1 or TIE2 as shown in **C** and **D**. (**A-B**) All comparisons were rendered non-significant by *two-tailed Mann-Whitney U test*.

The gene expression profiling of lung EC with neoadjuvant treatment with AB-Tie1-39 illustrated no significant changes in gene expression of either Tie1 or Tie2 (Fig. 3A). Next, we treated *in vitro* cultured HUVEC with AB-Tie1-39 or control-IgG for 24 h. Consistent with *in vivo* analysis, qPCR analysis of treated HUVEC revealed no major changes in gene expression of both, Tie1 and Tie2 (Fig. 3B). Additionally, immunoprecipitation-based analysis of soluble Tie1 and Tie2 from condition media showed no changes in protein shedding following antibody treatment. Overall, the data suggest that treatment with AB-Tie1-39, via inducing Tie2-mediated EC quiescence, promoted vascular stabilization without overtly affecting gene expression and surface presentation of TIE receptors.

COMMENT 2: AB-Tie1-39 specificity - Presumably the Ab does not recognize Tie2, as their ECDs are rather divergent, but was this tested?

RESPONSE 2: As mentioned by the reviewer, the extracellular domains of Tie1 and Tie2 are quite divergent. Nevertheless, employing HEK293 cells with overexpression of either Tie1 or Tie2, we show below that while AB-Tie1-39 failed to bind to Tie2, it specifically immunoprecipitated Tie1 (Fig. 4).

Figure 4. GFP, Tie1 or Tie2 was transient overexpressed HEK293 cells. Cell lysates were immunoprecipitated with AB-Tie1-39 and immunoblotted with either anti-TIE1 (on the left) or anti-TIE2 (on the right).

COMMENT 3: Effect of AB-Tie1-39 on EC apoptosis - Although the contextual effects of the Ab may be quite different in vitro vs. in vivo, does it have any effect on EC viability in ECs in vitro?

RESPONSE 3: AB-Tie1-39 conditioned HUVEC showed no differences in either Annexin V-based apoptosis or EdU-based proliferation assays (Fig. 5).

Figure 5. HUVEC were preconditioned with EdU and AB-Tie1-39 or IgG for 24 h. Subsequently, FACS-based analysis was performed with Annexin-V and EdU staining kits. Both comparisons were rendered non-significant by *two-tailed Mann-Whitney U test*.

COMMENT 4: Adjuvant therapy - Have the authors tested the Ab as a true adjuvant to chemotherapy? Although this may be beyond the scope of the current manuscript, as the data demonstrate that the Ab clearly has independent effects, it might be interesting to know how much of an additive effect the Ab confers.

RESPONSE 4: We thank the reviewer for bringing this clinically-important issue. We combined neoadjuvant AB-Tie1-39 with a single dose of maximum-tolerated chemotherapy (Abraxane, PTX) in the 4T1 metastasis model. Considering the strong suppression of distant metastases upon AB-Tie1-39 treatment, we had anticipated that it might be technically challenging to assess additive or synergistic survival benefit with combinatorial approaches. Indeed, combining neoadjuvant AB-Tie1-39 with MTD chemotherapy failed to provide additive benefit for the survival of mice (**Fig. 1**, please see also response to Reviewer 1, comments 1 and 4). Interestingly, although not significant, the combination group did even slightly worse as compared to AB-Tie1-39 single arm, which suggests that potential toxicity of chemotherapy to the lung vasculature might have compromised the vessel stabilization effects of AB-Tie1-39. We plan to follow this finding up in future studies.

COMMENT 5: Introduction, p. 3, paragraph 1 - The statement that, "... this gain translates in absolute numbers in an increase of OS within weeks to months" is confusing. "Within" implies that the improvement occurs quickly, whereas I believe the authors mean to say "an increase in OS of only weeks to months".

RESPONSE 5: We have corrected the text according to the reviewer's suggestion.

COMMENT 6: Fig. EV2A and B - Please define how tip cells were identified.

RESPONSE 6: Endothelial cells at the retinal front with filopodial extensions were defined as tip cells. We manually counted tip cells in 20x whole retina images and normalized the count to radial length of the vascular front. We have included the analysis details in the methods section of the revised manuscript.

COMMENT 7: Dose response - How was the Ab dose chosen, and was a dose response tested?

RESPONSE 7: Different anti-angiogenic drugs are frequently used in the range of 20 to 40 mg/kg. As AB-Tie1-39 is a humanized anti-human TIE1-binding antibody. We therefore opted for a higher dose of 40 mg/Kg for our *in vivo* experiments.

References

- Augustin HG, Koh GY, Thurston G, Alitalo K (2009) Control of vascular morphogenesis and homeostasis through the angiopoietin-Tie system. *Nat Rev Mol Cell Biol* 10: 165-77
- D'Amico G, Korhonen EA, Anisimov A, Zarkada G, Holopainen T, Hagerling R, Kiefer F, Eklund L, Sormunen R, Elamaa H, Brekken RA, Adams RH, Koh GY, Saharinen P, Alitalo K (2014) Tie1 deletion inhibits tumor growth and improves angiopoietin antagonist therapy. *J Clin Invest* 124: 824-34
- La Porta S, Roth L, Singhal M, Mogler C, Spegg C, Schieb B, Qu X, Adams RH, Baldwin HS, Savant S, Augustin HG (2018) Endothelial Tie1-mediated angiogenesis and vascular abnormalization promote tumor progression and metastasis. *J Clin Invest* 128: 834-845
- Saharinen P, Eklund L, Alitalo K (2017) Therapeutic targeting of the angiopoietin-TIE pathway. *Nat Rev Drug Discov* 16: 635-661
- Savant S, La Porta S, Budnik A, Busch K, Hu J, Tisch N, Korn C, Valls AF, Benest AV, Terhardt D, Qu X, Adams RH, Baldwin HS, Ruiz de Almodovar C, Rodewald HR, Augustin HG (2015) The orphan receptor Tie1 controls angiogenesis and vascular remodeling by differentially regulating Tie2 in tip and stalk cells. *Cell Rep* 12: 1761-73

Thank you for the submission of your revised manuscript to EMBO Molecular Medicine. We have now received the enclosed reports from the referees that were asked to re-assess it. As you will see the reviewers are now globally supportive and I am pleased to inform you that we will be able to accept your manuscript pending the following final amendments:

1) Please address the minor changes commented by referee 2.
Please provide a point-by-point letter INCLUDING my comments as well as the reviewer's reports and your detailed responses to their comments (as Word file).

***** Reviewer's comments *****

Referee #1 (Remarks for Author):

I was very positive about this manuscript when I initially reviewed it. It is even better now, and I believe the authors have satisfactorily addressed the questions or comments that I raised originally. This is a potentially important study that has ramifications for perioperative cancer surgery/therapy. One observation that I found intriguing and should be followed up in a subsequent study is the fact that the Tie-1 antibody when administered with taxane based chemotherapy actually resulted in a less effective survival outcome compared to the Tie-1 antibody alone. The authors have speculated why this may have happened. I think this deserves further experimental scrutiny since there is a tendency in medical oncology to test a new therapeutic by adding it to an approved standard-of-care therapy, when doing neoadjuvant, adjuvant, or metastatic therapy. Here, the results argue for not adopting this approach and it would be helpful for possible future clinical evaluation to have a better understanding about why the Tie-1 activating (in vivo) antibody did not result in a superior benefit when combining it with conventional chemotherapy.

Referee #2 (Remarks for Author):

Comment to response 1

The apparent discrepancy between genetic and pharmacological targeting of Tie1 is not completely addressed. In figure 3D it is possible to see the accumulation of AB-Tie1-39 in lung lysates but not in primary tumors. The authors should show the amount of antibody at the primary tumor in order to compare with the amount in the lung. Furthermore, about the TAMs issue, it cannot be excluded that Tie expression in TAMs changes in the timeframe of the treatment or in the different stages of tumor progression. The authors should address this question. The authors should also make explicit the percentage of M1-polarized antitumor TAMs (defined as percentage of CD206low/CD11chigh cells on F4/80+ macrophages) and the percentage of protumor M2 TAMs (defined as CD206high/CD11clow cells on F4/80+ macrophages).

Comment to response 2

The authors explain the effect of AB-Tie1-39 as restricted to the extravasation thus blocking circulating tumor cells. Is it possible speculate on the effect of this treatment on metastasis regression and on disease stabilization in a metachronous model (when metastasis is already established)? This aspect should be discussed.

There are no additional comments, all the other questions have been fully addressed.

Referee #3 (Comments on Novelty/Model System for Author):

The revised manuscript has improved technical details with strong statistical analysis. The novelty of targeting Tie1 is high, as there are no approaches to do this, inspire of data showing Tie1 is required for tumor angiogenesis. The inability to affect tumor size while limiting tumor metastasis slightly reduces the paper's medical impact. The tumor models used are appropriate for the study.

Referee #3 (Remarks for Author):

The authors have adequately addressed my prior concerns

2nd Revision - authors' response

16th Mar 2020

Reviewer #1

COMMENT1: I was very positive about this manuscript when I initially reviewed it. It is even better now, and I believe the authors have satisfactorily addressed the questions or comments that I raised originally. This is a potentially important study that has ramifications for perioperative cancer surgery/therapy. One observation that I found intriguing and should be followed up in a subsequent study is the fact that the Tie-1 antibody when administered with taxane based chemotherapy actually resulted in a less effective survival outcome compared to the Tie-1 antibody alone. The authors have speculated why this may have happened. I think this deserves further experimental scrutiny since there is a tendency in medical oncology to test a new therapeutic by adding it to an approved standard-of-care therapy, when doing neoadjuvant, adjuvant, or metastatic therapy. Here, the results argue for not adopting this approach and it would be helpful for possible future clinical evaluation to have a better understanding about why the Tie-1 activating (in vivo) antibody did not result in a superior benefit when combining it with conventional chemotherapy.

RESPONSE TO GENERAL COMMENT: We sincerely appreciate the reviewer's positive assessment of the manuscript. Indeed, we too have been intrigued by the magnitude of anti-metastatic phenotypes that mono-therapeutic administration of AB-Tie1-39 offered in our preclinical experiments. Further, vessel stabilization at the metastatic site established AB-Tie1-39 as a potent anti-migrastatic agent. Similar to the reviewer, we did not anticipate that the addition of chemotherapy would not be synergistic to AB-Tie1-39 monotherapy, and these unexpected findings are being followed up in separate experiments. We strongly believe that the data presented here argue for anti-metastatic therapies to be assessed differently than classical anti-cancer drugs and for the mechanism-based combination of available therapeutic options in the future.
Singhal, Gengenbacher, La Porta et al. EMM-2019-111643

Reviewer #2

COMMENT1: The apparent discrepancy between genetic and pharmacological targeting of Tie1 is not completely addressed. In figure 3D it is possible to see the accumulation of AB-Tie1-39 in lung lysates but not in primary tumors. The authors should show the amount of antibody at the primary tumor in order to compare with the amount in the lung. Furthermore, about the TAMs issue, it cannot be excluded that Tie expression in TAMs changes in the timeframe of the treatment or in the different stages of tumor progression. The authors should address this question. The authors should also make explicit the percentage of M1-polarized antitumor TAMs (defined as percentage of CD206low/CD11chigh cells on F4/80+ macrophages) and the percentage of protumor M2 TAMs (defined as CD206high/CD11clow cells on F4/80+ macrophages).

COMMENT1: We thank the reviewer for his/her thoughtful review of our manuscript. In the present study, we focused primarily on (i) developing and establishing a novel Tie1 function-blocking antibody AB-Tie1-39, and (ii) assessing the therapeutic potential of AB-Tie1-39 on metastatic progression. Perioperative administration of monotherapy AB-Tie1-39 prolonged mouse survival without altering primary tumors as well as in a reductionist, yet mechanistically-revealing, experimental metastasis assay, AB-Tie1-39 strongly impeded extravasation of intravenously-injected tumor cells into the lungs, thereby underlining a crucial impact of Tie1-blocking on vessel stabilization at the metastatic site. We agree with the reviewer that a clearer understanding of the discrepancy between genetic and pharmacological inhibition of Tie1 will be important for the clinical development of Tie1 as a therapeutic target, and we are following up on these observations to further understand the molecular mechanisms including the impact of Tie1-blocking on myelopoiesis. We have modified Fig. EV5F according to the reviewer's suggestion.

COMMENT 2: The authors explain the effect of AB-Tie1-39 as restricted to the extravasation thus blocking circulating tumor cells. Is it possible speculate on the effect of this treatment on metastasis

regression and on disease stabilization in a metachronous model (when metastasis is already established)? This aspect should be discussed.

RESPONSE 2: We have performed postsurgical adjuvant administration of AB-Tie1-39 in two independent preclinical metastasis models (LLC and 4T1), in which therapy was applied one day after the resection of primary tumors. Subsequently, mice developed lung and lymph node metastases in the absence of a primary tumor, suggesting that tumor cells had already disseminated and seeded at the metastatic site prior to therapy initiation. Coherently in both LLC and 4T1 models, postsurgical adjuvant administration of AB-Tie1-39 failed to improve the overall survival of mice as compared to IgG-treated cohort of mice. Therefore, the data imply that AB-Tie1-39 is potentially not efficacious for the regression of already growing metastasis but exhibits an anti-migrastatic effect by restricting extravasation of circulating tumor cells.

Reviewer #3

COMMENT 1: The revised manuscript has improved technical details with strong statistical analysis. The novelty of targeting Tie1 is high, as there are no approaches to do this, inspire of data showing Tie1 is required for tumor angiogenesis. The inability to affect tumor size while limiting tumor metastasis slightly reduces the paper's medical impact. The tumor models used are appropriate for the study. The authors have adequately addressed my prior concerns.

COMMENT 1:

We sincerely appreciate reviewer's positive assessment of our manuscript.

Corresponding Author Name: Hellmut G. Augustin

Manuscript Number: EMM-2019-11164